# CDEA: Context- and Detail-Enhanced Unsupervised Learning for Domain Adaptive Semantic Segmentation

Shuyuan Wen*
School of Instrument Science and
Optoelectronic Engineering &
Shen Yuan Honors College,
Beihang University
Beijing, China
shuyuanwen@buaa.edu.cn

Bingrui Hu
School of Instrument Science and
Optoelectronic Engineering,
Beihang University
Beijing, China
bingruihu@buaa.edu.cn

Wenchao Li
School of Instrument Science and
Optoelectronic Engineering,
Beihang University
Beijing, China
wenchaoli@buaa.edu.cn

## Abstract

Unsupervised domain adaptation (UDA) aims to adapt a model trained on the source domain (e.g. synthetic data) to the target domain (e.g. real-world data) without requiring further annotations on the target domain. Most previous UDA methods for semantic segmentation focus on minimizing the domain discrepancy of various levels, e.g., pixels and features, for extracting domain-invariant knowledge. However, the primary domain knowledge, such as context and detail correlation, remains underexplored. To address this problem, we propose a context- and detail-enhanced unsupervised learning framework, called CDEA, for domain adaptive semantic segmentation that facilitates image detail correlations and contexts semantic consistency. Firstly, we propose an adaptive masked image consistency module to enhance UDA by learning spatial context relations of the target domain, which enforces the consistency between predictions and masked target images. Secondly, we propose a detail extraction module to enhance UDA by integrating the learning of spatial information into low-level layers, which fuses the low-level detail features with deep semantic features. Extensive experiments verify the effectiveness of the proposed method and demonstrate the superiority of our approach over state-of-the-art methods.

## CCS Concepts

• **Computing methodologies** → **Scene understanding**; **Transfer learning**; • **Networks** → **Network algorithms**.

## Keywords

Context-Enhanced, Detail-Enhanced, Domain Adaptive, Unsupervised Learning, Semantic Segmentation

**ACM Reference Format:**
Shuyuan Wen, Bingrui Hu, and Wenchao Li. 2024. CDEA: Context- and Detail-Enhanced Unsupervised Learning for Domain Adaptive Semantic Segmentation. In *Proceedings of the 32nd ACM International Conference on Multimedia (MM '24), October 28-November 1, 2024, Melbourne, VIC, Australia.* ACM, New York, NY, USA, 9 pages. https://doi.org/10.1145/3664647.3681111

---

*Corresponding author.

---

## 1 Introduction

Deep neural networks have achieved significant success in recent years across various multimedia tasks [1], such as cross-modal retrieval, image captioning, etc. However, their training often requires a large amount of annotated data. Providing annotations is particularly labor-intensive for specific tasks, such as semantic segmentation [2]. Besides, pixel-wise labels of the entire image are necessary, and it can take more than one hour per image [3]. However, a network trained on such a source dataset usually performs worse when applied to the target dataset [4], as neural networks are sensitive to domain gaps. To minimize such a gap, researchers resort to Unsupervised Domain Adaptation (UDA) to transfer the knowledge from labeled source-domain data to the unlabeled target-domain environment.

Domain adaptive semantic segmentation has got great attention and various methods have been proposed in the last few years, which can be roughly divided into two categories: adversarial training [5–7] and unsupervised training [8–10]. However, there is still a noticeable performance gap compared to supervised training [5]. A common challenge is the confusion of classes with a similar visual appearance on the target domain, such as "wall/fence" or "pedestrian/rider", The reason is that no ground truth supervision is available to learn the slight appearance differences. For example, the segmentation class of the fence in Fig. 1(a) is segmented as a wall, probably as a result of a similar local look. In addition, we find that the edge of category prediction results is not significant and exists a large noise.

To address this challenge, we propose to enhance UDA with spatial context relations as additional clues for robust visual recognition. For instance, the curb in the foreground of Fig. 1(a) could be a crucial context relation to recognizing the fence despite the ambiguous texture correctly. Hoyer *et al.* [11] propose a mask patch method by learning spatial information of the target domain, which achieves robust visual recognition. Fan *et al.* [12] propose a short-term dense concatenate network to learn detailed information, which improves edge extraction capability. Although the used network architectures already can model context relations, previous UDA methods can still not reach the full potential of using context dependencies on the target domain as the unsupervised target losses are not powerful enough to enable effective learning of such information.

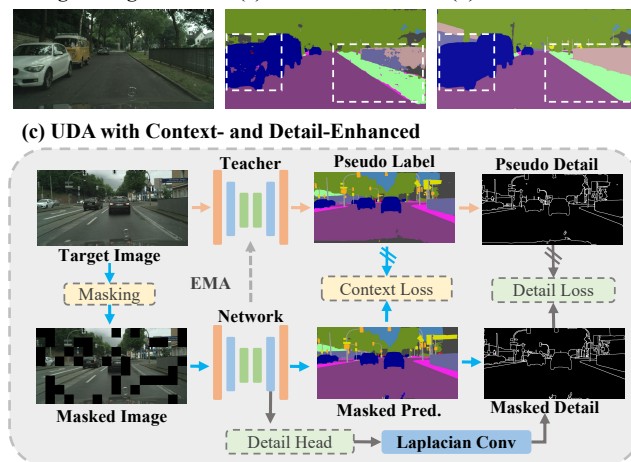

**Figure 1: (a) Previous UDA methods such as HRDA struggle with similar appearance classes on the target domain. (b) The proposed Context- and Detail-Enhanced (CDEA) has solved the above challenges to enhance the learning of context and detail relations. With CDEA, the adapted network can correctly segment the wall and sidewalk. (c) Different from existing works, we focus on learning the intra-domain information and argue that the contextual structure between detail and mask patches can facilitate the model learning the domain-invariant knowledge in an unsupervised manner.**

Therefore, we design a method to explicitly encourage the network to learn comprehensive context relations of the target domain during UDA. In particular, we propose a novel Context- and Detail-Enhanced (CDEA) plug-in for UDA, shown as Fig. 1(c), which can be applied to various visual recognition tasks. Considering semantic segmentation for illustration, CDEA masks out a random selection of target image patches and trains the network to predict the semantic segmentation result of the entire image, including the masked regions. Therefore, the network has to utilize the context to infer the semantics of the masked-out regions, which can increase the robustness of the network.

As there are no ground truth labels for the target domain, we resort to pseudo-labels, generated by an EMA teacher [13] that uses the original, unmasked target images as input. Therefore, the teacher can utilize both context and local knowledge to generate robust pseudo-labels. During the training, different classes of objects are masked out so that the network learns to utilize different context knowledge, which further increases the robustness [11]. After UDA with Context- and Detail-Enhanced, the network can better exploit context knowledge and correctly segment difficult areas that rely on context knowledge, such as the sidewalk in Fig. 1(b).

To the best of our knowledge, CDEA is the first UDA method to fuse masked images and edge features to facilitate learning context and detail relations on the target domain. Due to its universality and simplicity, Context- and Detail-Enhanced can be straightforwardly integrated into various UDA methods across multiple visual recognition tasks, making it highly valuable in practice. CDEA achieves significant and consistent performance improvements for different

UDA methods, including adversarial training and unsupervised training on semantic segmentation tasks with domain gaps and network architectures. The main contributions of this paper can be summarized as follows:

(1) Different from existing works on semantic segmentation, we focus on mining domain-invariant knowledge from the original domain in an unsupervised manner. We propose a context- and detail-enhanced unsupervised learning framework to harness both context- and detail-wise consistency against different contexts, which is well-aligned with the segmentation task.

(2) Our unsupervised learning method does not require extra annotations and is compatible with other existing UDA frameworks. The effectiveness of CDEA has been tested by extensive ablation studies, and it achieves competitive accuracy on two commonly used UDA benchmarks, 76.3 mIoU on GTA→Cityscapes and 68.4 mIoU on Synthia→Cityscapes.

(3) We achieve a new state-of-the-art (SOTA), and outperform other unsupervised domain adaptive methods by a large margin in solving both context and detail-enhanced problems for semantic segmentation.

## 2 Related Work
### 2.1 Unsupervised Domain Adaptation

In recent years, semantic segmentation approaches have been based on deep neural networks, which can be effectively trained in an end-to-end manner to perform pixel-wise classification [14–17]. However, semantic segmentation approach training often requires a large amount of annotated data, which can take more than one hour per image. Unsupervised domain adaptation (UDA) enables a model to transfer scalable knowledge from a label-rich source domain to a label-scarce target domain, which improves the adaptation performance. Several strategies have been proposed to adapt a semantic segmentation network to the target domain, which can be grouped into adversarial training [5–7] and unsupervised training [8–10, 18].

Adversarial training methods aim to learn domain-invariant knowledge based on adversarial domain alignment. Wang *et al.* [19] propose a fine-grained adversarial learning strategy for class-level feature alignment while preserving the internal structure of semantics across domains. Shan *et al.* [7] propose to conduct an auxiliary adversarial training on the fused multi-level CNN features. However, unstable adversarial training methods usually lead to suboptimal performance. Unsupervised training methods aim to create pseudo labels for the target domain images using the model trained by labeled source domain data, and then the model is re-trained by pseudo labels. He *et al.* [20] build a dynamic dictionary with a queue and a moving-averaged encoder to build a large and consistent dictionary on-the-fly that facilitates contrastive unsupervised learning. Van *et al.* [21] adopts a predetermined mid-level prior in a contrastive optimization objective to learn pixel embeddings. Different from the above-mentioned works, we focus on learning the domain knowledge of context and detail in an unsupervised manner. The proposed method is complementary to the existing approach to further boost the result.

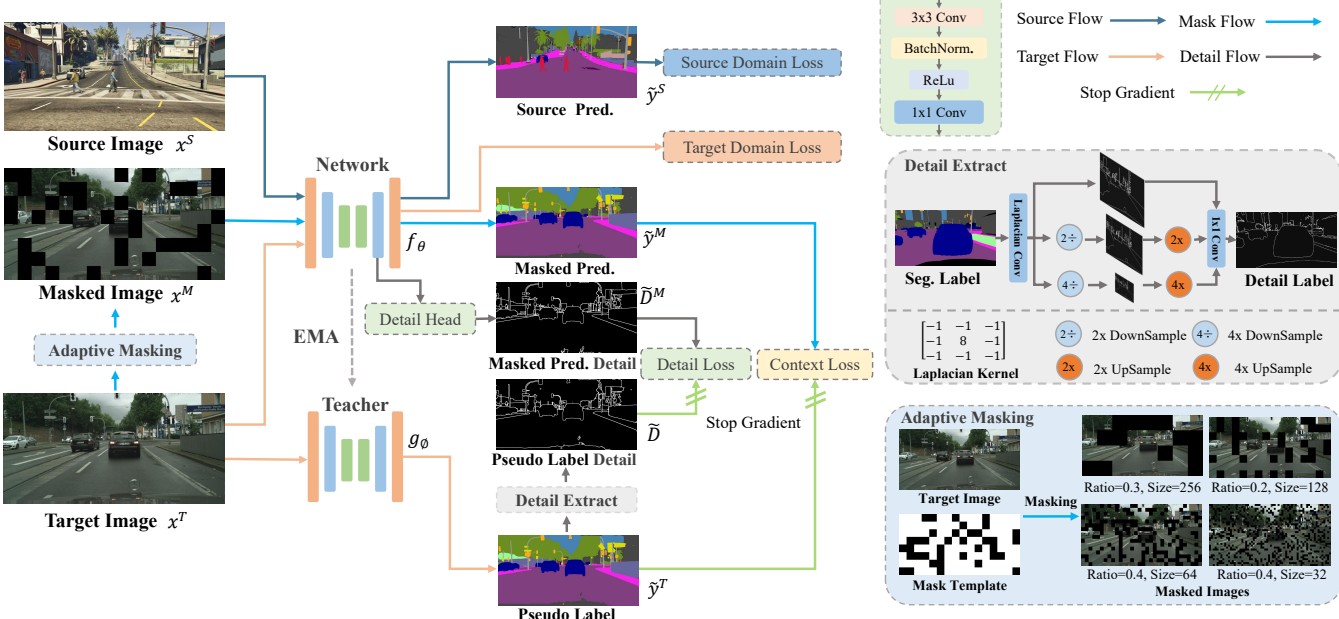

**Figure 2: A brief illustration of our Context- and Detail-Enhanced Unsupervised Learning Framework (CDEA), based on a teacher-student architecture. Given the labeled source data $\{(x^S, y^S)\}$, we utilize the network $f_\theta$ to calculate the segmentation prediction, supervised by the Source Domain loss $\mathcal{L}_k^S$. During training, we leverage the moving averaged model $g_\phi$ to generate the pseudo label $\tilde{y}^T$ to craft the masked label $\tilde{y}^M$. The $\tilde{D} \in \mathbb{R}^{H \times W}$ denotes the pseudo detail ground-truth and $\tilde{D}^M \in \mathbb{R}^{H \times W}$ denotes the masked predicted detail. According to the masked label, we copy the corresponding regions as the masked data $x^M$. We also deploy the network $g_\phi$ to obtain the masked prediction $\tilde{y}^M$ supervised by $\mathcal{L}_k^T$. Except for the above-mentioned basic segmentation losses, we propose Context Loss $\mathcal{L}_k^M$ and Detail Loss $\mathcal{L}_k^D$. During inference, we drop the Detail Head $H_{Detail}$ and only keep the network $f_\theta$.**

## 2.2 Context and Detail Enhancement

Several UDA methods propose network components, such as cross-domain attention [7], or context-aware feature fusion [22] can capture contexts to facilitate learning domain-robust context dependencies. Although these network modules can capture context, the unsupervised loss on the target domain does not provide sufficient supervision to learn all relevant target context and detail relations. For example, the segmentation performs poorly in details and edges as a result of a similar local look, shown in Fig. 1(a). Research has shown that the domain knowledge of context and detail Has a significant impact on performance in semantic segmentation [23]. Therefore, we focus on learning the domain knowledge of context and detail in an unsupervised manner.

Recently, semantic segmentation approaches have been based on deep neural networks. Hoyer *et al.* [11] propose a mask patch method by learning spatial information of the target domain, which achieves robust visual recognition. However, these mechanisms can not capture all relevant context information and the prediction results perform poorly in details and edges. Fan *et al.* [12] propose a short-term dense concatenate network to learn detailed information, which improves edge extraction capability. Different from the above-mentioned works [11], The proposed Context- and Detail-Enhanced can learn more extensive multi-scale context information due to the random size of patch masking during training. Moreover, we introduce a detail-enhanced module in an unsupervised manner according

to [12], which can improve the segmentation performance in details and edges. Therefore, the multi-grained unsupervised learning on both context and detail are complementary to each other and can learn the domain-invariant context feature.

## 3 Approach

In this section, We first introduce the UDA definition and segmentation losses for semantic segmentation domain adaptation in Sec. 3.1. Then, we introduce an adaptive masked image consistency module, which enforces the consistency between predictions and masked target images in Sec. 3.2. Besides, we introduce the detail-enhanced module, leading to more precise preservation of spatial details in low-level layers in Sec. 3.3. We finally discuss the mechanism of the proposed method in Sec. 3.4. Our proposed method is designed to be applicable to common network architectures and can be combined with existing UDA methods.

## 3.1 Unsupervised Domain Adaptation

The UDA neural network can be trained on the source domain synthetic data $X^S = \{x_k^S\}_{k=1}^{N_S}$ labeled by $Y^S = \{y_k^S\}_{k=1}^{N_S}$ and the unlabelled target domain real-world data $X^T = \{x_k^T\}_{k=1}^{N_T}$ with a supervised source loss and adaptation target loss, where $N_S$ and $N_T$ are the numbers of images in the source and target domain, respectively. Domain adaptive semantic segmentation intends to

learn a mapping function that projects the input image $x^T$ to the segmentation prediction $\tilde{y}^T$ in the target domain.

**Source Domain Losses.** The specific source loss depends on the computer vision task. For semantic segmentation, the pixel-wise cross-entropy is typically used.

$$\mathcal{H}(\hat{y}, y) = -\sum_{i=1}^{H}\sum_{j=1}^{W}\sum_{c=1}^{C} p_{ijc} \log \hat{y}_{ijc} \tag{1}$$

where $H, W$ is the height and width of the input image. The label $y^S$ belongs to $C$ categories. $p_{ijc}$ is the one-hot vector of label $y_{ijc}$, and the value $p_{ijc}(c)$ equals to 1 if $c == y_{ijc}$ otherwise 0.

We learn the basic source domain knowledge by adopting the segmentation loss on the source domain, which can be formulated as:

$$\mathcal{L}_k^S = \mathcal{H}(f_\theta(x_k^S), y_k^S) \tag{2}$$

where we utilize the visual backbone and 2-layer Multi-Layer Perceptrons (MLPs) as $f_\theta$ for segmentation category prediction.

**Target Domain Loss.** However, a model trained on the source domain usually experiences a performance drop when applied to another domain. Considering that there are no labels for the target-domain data, we generate pseudo labels $\tilde{Y}^T = \{\tilde{y}_k^T\}$ for the target domain data $X^T$ by a teacher network $g_\phi$ during training to learn the knowledge from the target domain, where $\tilde{y}_k^T = argmax(g_\phi(x_k^T))$. Therefore, UDA methods can use unlabeled images from the target domain $X^T = \{x_k^T\}_{k=1}^{N_T}$ to adapt the network, the segmentation loss on the target domain can be formulated as:

$$\mathcal{L}_k^T = \mathcal{H}(f_\theta(x_k^T), \tilde{y}_k^T) \tag{3}$$

where $\tilde{y}_k^T$ is the pseudo label of $x_k^T$.

In practice, the teacher network $g_\phi$ is set as the Exponential Moving Average (EMA) of the weights of the student network $f_\theta$ after each training iteration. Therefore, the unsupervised loss for the target domain $\mathcal{L}^T$ is added to the optimization problem with a weight $\lambda^T$

$$\min_\theta \frac{1}{N_S}\sum_{k=1}^{N_S} \mathcal{L}_k^S + \frac{1}{N_T}\sum_{k=1}^{N_T} \lambda^T \mathcal{L}_k^T . \tag{4}$$

## 3.2 Context Enhance of High-level Features

Recently, many network architectures can integrate both local and context information in their features to recognize an object [22, 24]. Although the learning of context relations can be guided by ground truth in supervised learning, without ground truth supervision is available for the target domain in UDA. However, the existing unsupervised losses are not consequential enough to enable effective learning of context information, such as in Fig. 1.

Therefore, we propose encouraging the learning performance of context relations on the target domain to provide context information for robust recognition of classes with similar local appearances. To improve the learning context relations in the target domain, we introduce a Context Enhance Module (CEM), which can be easily plugged into various existing UDA methods. The domain adaptation process with the CEM is illustrated in Fig. 3 and explained below. The CEM withholds local information by randomly masking out

---

**Algorithm 1** CDEA algorithm

**Input:** Source domain data $X^S$ with labels $Y^S$, Target domain data $X^T$, segmentation network $f_\theta$ with detail head $H_{\text{Detail}}$, segmentation teacher network $g_\phi$, the total iteration number $T_{\text{total}}$.

1: Initialize network parameter $\theta$ with ImageNet pre-trained parameters. Initialize teacher network $\phi$ randomly.
2: **for** iteration = $1 \to T_{\text{total}}$ **do**
3:     $x^S, y^S \sim U., x^T \sim V$.
4:     $x_M^T \leftarrow Mask(x^T, r, s)$. Generate masked image from param mask-ratio $r$ and mask-size $s$.
5:     Generate pseudo labels. $\tilde{y}^T \leftarrow argmax(g_\phi(x^T))$.
6:     Compute predictions. $\hat{y}^S \leftarrow argmax(f_\theta(x^S))$.
7:     Compute detail. $\tilde{D} \leftarrow argmax(Laplacian(f_\theta(x^T)))$, $\hat{D}^M \leftarrow argmax(H_{Detail}(f_\theta(x_M^T)))$.
8:     Compute Loss for the mini-batch $\mathcal{L}_{\text{total}}$.
9:     Compute $\nabla_\theta \mathcal{L}_{\text{total}}$ by backpropagation.
10:     Perform stochastic gradient descent.
11:     Update teacher network $\phi$ with $\theta$.
12: **end for**
13: **return** Segmentation network $f_\theta$.

---

patches of the target image. Therefore, a patch mask M is randomly sampled from a uniform distribution

$$\mathcal{M}(x, y) = [v > r], \text{ if } \begin{cases} x \in [ms+1 : ms+s], \\ y \in [ns+1 : ns+s] \end{cases} \tag{5}$$

where $v \sim \mathcal{U}(0, 1)$, $s$ is the mask-size, $r$ is the mask-ratio, $m \in [0 \dots W/s - 1]$, $n \in [0 \dots H/s - 1]$ the patch indices, and $[\cdot]$ denotes the Iverson bracket.

$$x_M^T = \mathcal{M} \odot x^T \tag{6}$$

where the target image $x^T$. The masked target image $x_M^T$ is obtained by element-wise multiplication $\odot$ of mask and image, shown as Fig. 3.

The masked target prediction $\hat{y}_M^T$ can use the unmasked image regions' limited information to learn the context information. Therefore, we use the remaining context information to reconstruct the correct label without access to the entire image, the Context Enhance loss $\mathcal{L}^M$ is introduced

$$\mathcal{L}^M = \mu \mathcal{H}(f_\theta(x_M^T), \tilde{y}^T) \tag{7}$$

where $\tilde{y}^T$ is a pseudo-label and $\lambda^T$ is quality weight. The pseudo-label $\tilde{y}^T$ is the prediction of a teacher network $g_\phi$ of the complete target image $x^T$. The pixel-wise cross-entropy $\mathcal{H}(\cdot)$ be shown in Eq. (1).

The existing study found that there are potential faults in pseudo-labels, especially at the beginning of the training [11]. Therefore, we design a weight of quality estimate $\lambda^T$ for the Context Enhance loss and utilize the ratio of pixels exceeding a threshold $\tau$ of the maximum softmax probability.

$$\mu = \frac{1}{H \cdot W}\sum_{i=1}^{H}\sum_{j=1}^{W}\sum_{i=1}^{C}[\max_c g_\phi(x_{ijc}^T) > \tau]. \tag{8}$$

## 3.3 Detail Enhance of Low-level Features

Although existing network architectures can already model detailed information in supervised learning, previous UDA methods can still not reach the full potential of using detailed dependencies on the target domain. As shown in Fig. 1, the existing unsupervised methods easily neglect the detailed information about boundaries because they do not learn detailed information effectively. Therefore, we propose a Detail Enhance Module (DEM) to manually guide the low-level layers in learning the spatial details.

**Detail Head:** Compared with the high-level semantic features, the low-level features of neural networks can encode more spatial details [25]. Thus, we insert the Detail Head into the low-level backbone layer to generate the detailed feature map, shown in Fig. 2. We first generate the detailed feature map ground truth from the segmentation ground truth by the Laplacian operator. Then, we use the detailed ground truth as the guide of a detailed feature map to guide the low-level layers in learning the features of spatial details. Finally, the learned detail features are fused with the context features from the deep block of the decoder for segmentation prediction. As shown in Fig. 2, the Detail Extract introduces the generating process of detail ground truth.

We use the Detail Head to produce the detailed map, which guides the low-level layer to encode spatial information, as shown in Fig. 2. The Detail Head includes a $3 \times 3$ Conv-BN-ReLU operator followed by a $1 \times 1$ convolution to get the output detail map. Then, we use a bilinear interpolation to up-sample the feature map to the size of ground truth. In the experiment, the Detail Head effectively enhances the feature representation. Therefore, this detail-information can easily boost the accuracy of the segmentation task without any cost in inference.

**Detail Enhance Loss:** We model the detail prediction as a binary segmentation task. We generate the binary detail ground truth from the semantic segmentation ground truth. Considering the number of detail pixels is much less than the non-detail pixels, detail prediction is a class-imbalance problem, which often uses weighted cross-entropy to solve [26]. However, the weighted cross-entropy always leads to coarse results. Dice loss [27] is insensitive to the number of foreground/background pixels, which can measure the overlap between predicted maps and ground truth, thus can alleviate the class imbalance problem [28].

Therefore, we adopt binary cross-entropy $\mathcal{H}(\cdot)$ and Dice loss $\mathcal{L}_{Dice}$ to jointly optimize detail learning. So for the predicted detail map with height $H$ and width $W$, the Detail loss $\mathcal{L}_{Detail}$ is formulated as follows:

$$\mathcal{L}_{Detail}(\tilde{D}, \tilde{D}^M) = \mathcal{H}(\tilde{D}, \tilde{D}^M) + \mathcal{L}_{Dice}(\tilde{D}, \tilde{D}^M) \tag{9}$$

where $\tilde{D} \in \mathbb{R}^{H \times W}$ denotes the pseudo detail ground-truth and $\tilde{D}^M \in \mathbb{R}^{H \times W}$ denotes the masked predicted detail. The binary cross-entropy $\mathcal{H}(\cdot)$ be shown in Eq. (1). The Dice loss $\mathcal{L}_{Dice}$ is given as follows:

$$\mathcal{L}_{Dice}(\tilde{D}, \tilde{D}^M) = 1 - \frac{2\sum_1^{H \times W}(\tilde{D}_i \tilde{D}_i^M) + \epsilon}{\sum_1^{H \times W}(\tilde{D}_i)^2 + \sum_1^{H \times W}(\tilde{D}_i^M)^2 + \epsilon} \tag{10}$$

where $i$ denotes the $i$-th pixel and $\epsilon$ is a Laplace smoothing item to avoid zero division. In this paper, we set $\epsilon = 1$.

## 3.4 Method Summary and Discussion

**Total Loss.** The teacher network $g_\phi$ is implemented as an Exponential Moving Average (EMA) teacher. It is a common strategy used in UDA [13]. Its weights are the EMA of the weights of $f_\theta$ with smoothing factor $\alpha$

$$\phi_{t+1} \leftarrow \alpha\phi_t + (1 - \alpha)\theta_t \tag{11}$$

where $t$ is a training step. The EMA teacher realizes a temporal ensemble of previous student models $f_\theta$, which increases the robustness and temporal stability of pseudo-labels. As the teacher is updated based on the student $f_\theta$, it will gradually obtain the context enhanced learning capability from $f_\theta$. In contrast to the student $f_\theta$, the teacher $g_\phi$ has privileged access to the original image $x^T$ (see Eq. 8). Thus, it can utilize the context and the detial information to generate higher-quality pseudo-labels.

The overall training objective is the combination of pixel-level cross-entropy loss and the proposed CDEA:

$$\min_\theta \frac{1}{N_S} \sum_{k=1}^{N_S} \mathcal{L}_k^S + \frac{1}{N_T} \sum_{k=1}^{N_T} (\lambda^T \mathcal{L}_k^T + \lambda^M \mathcal{L}_k^M + \lambda^D \mathcal{L}_k^D). \tag{12}$$

We summarize the overall training process of our CDEA framework in Algorithm 1.

**Discussion. 1. Correlation between Context and Detail Enhance.** Both context and detail enhance are derived from different levels of learning, and they work at different effect regions, pixel-wise and patch-wise. However, detail enhance explores the correlation of different edge categories over the whole image, while context enhance imposes regularization on the mask patches from a local semantic perspective. Therefore, the two enhanced methods are complementary and can learn the intra-domain inherent context and detail within the data.

**2. What is the advantage of the proposed approach?** Previous UDA methods mainly focused on designing the module for feature learning in the target data. Differently, we are motivated by the objectives of UDA semantic segmentation in a manner and thus leverage context and detail correlations in the target data to facilitate key features and knowledge learning. By explicitly regularizing the feature space via CDEA, we enable the model to explore the inherent intra-domain context and detail information in unsupervised learning, pixel-wise and patch-wise, without extra parameters or annotations. Therefore, CDEA can be easily integrated into existing UDA approaches to achieve better results without extra work during inference.

**3. Difference from semantic segmentation for UDA.** Conventional contrastive learning methods tend to perform contrast in the instance or pixel level alone. We formulate context and detail enhance in a similar format but focus on the local effect regions within the images, which aligns well with the local-focused segmentation task. We show that the proposed context and detail enhance, and regularize the domain adaptation training and guide the model to shed more light on the intra-domain context. The experiment result verifies our innovation that context and detail enhance and improve smooth edges between different categories and output a higher accuracy on small-object categories.

**Table 1: Quantitative results with previous UDA methods on GTA → Cityscapes. We present pre-class IoU and mIoU. The best accuracy in every column is in bold.**

| Method | Road | SW | Build | Wall | Fence | Pole | TL | TS | Veg. | Terrain | Sky | PR | Rider | Car | Truck | Bus | Train | Motor | Bike | mIoU |
|---|---|---|---|---|---|---|---|---|---|---|---|---|---|---|---|---|---|---|---|---|
| AdvEnt [29] | 89.4 | 33.1 | 81.0 | 26.6 | 26.8 | 27.2 | 33.5 | 24.7 | 83.9 | 36.7 | 78.8 | 58.7 | 30.5 | 84.8 | 38.5 | 44.5 | 1.7 | 31.6 | 32.4 | 45.5 |
| CyCADA [30] | 86.7 | 35.6 | 80.1 | 19.8 | 17.5 | 38.0 | 39.9 | 41.5 | 82.7 | 27.9 | 73.6 | 64.9 | 19.0 | 65.0 | 12.0 | 28.6 | 4.5 | 31.1 | 42.0 | 42.7 |
| CLAN [31] | 87.0 | 27.1 | 79.6 | 27.3 | 23.3 | 28.3 | 35.5 | 24.2 | 83.6 | 27.4 | 74.2 | 58.6 | 28.0 | 76.2 | 33.1 | 36.7 | 6.7 | 31.9 | 31.4 | 43.2 |
| AdaptSegNet [32] | 86.5 | 36.0 | 79.9 | 23.4 | 23.3 | 23.9 | 35.2 | 14.8 | 83.4 | 33.3 | 75.6 | 58.5 | 27.6 | 73.7 | 32.5 | 35.4 | 3.9 | 30.1 | 28.1 | 42.4 |
| SP-Adv [7] | 86.2 | 38.4 | 80.8 | 25.5 | 20.5 | 32.8 | 33.4 | 28.2 | 85.5 | 36.1 | 80.2 | 60.3 | 28.6 | 78.7 | 27.3 | 36.1 | 4.6 | 31.8 | 28.4 | 44.3 |
| ASA [33] | 89.2 | 27.8 | 81.3 | 25.3 | 22.7 | 28.7 | 36.5 | 19.6 | 83.8 | 31.4 | 77.1 | 59.2 | 29.8 | 84.3 | 33.2 | 45.6 | 16.9 | 34.5 | 30.8 | 45.1 |
| MaxSquare [34] | 88.1 | 27.7 | 80.8 | 28.7 | 19.8 | 24.9 | 34.0 | 17.8 | 83.6 | 34.7 | 76.0 | 58.6 | 28.6 | 84.1 | 37.8 | 43.1 | 7.2 | 32.3 | 34.2 | 44.3 |
| APODA [6] | 85.6 | 32.8 | 79.0 | 29.5 | 25.5 | 26.8 | 34.6 | 19.9 | 83.7 | 40.6 | 77.9 | 59.2 | 28.3 | 84.6 | 34.6 | 49.2 | 8.0 | 32.6 | 39.6 | 45.9 |
| MRNet [35] | 89.1 | 23.9 | 82.2 | 19.5 | 20.1 | 33.5 | 42.2 | 39.1 | 85.3 | 33.7 | 76.4 | 60.2 | 33.7 | 86.0 | 36.1 | 43.3 | 5.9 | 22.8 | 30.8 | 45.5 |
| APODA [6] | 85.6 | 32.8 | 79.0 | 29.5 | 25.5 | 26.8 | 34.6 | 19.9 | 83.7 | 40.6 | 77.9 | 59.2 | 28.3 | 84.6 | 34.6 | 49.2 | 8.0 | 32.6 | 39.6 | 45.9 |
| CBST [36] | 91.8 | 53.5 | 80.5 | 32.7 | 21.0 | 34.0 | 28.9 | 20.4 | 83.9 | 34.2 | 80.9 | 53.1 | 24.0 | 82.7 | 30.3 | 35.9 | 16.0 | 25.9 | 42.8 | 45.9 |
| PatchAlign [37] | 92.3 | 51.9 | 82.1 | 29.2 | 25.1 | 24.5 | 33.8 | 33.0 | 82.4 | 32.8 | 82.2 | 58.6 | 27.2 | 84.3 | 33.4 | 46.3 | 2.2 | 29.5 | 32.3 | 46.5 |
| BL [38] | 91.0 | 44.7 | 84.2 | 34.6 | 27.6 | 30.2 | 36.0 | 36.0 | 85.0 | 43.6 | 83.0 | 58.6 | 31.6 | 83.3 | 35.3 | 49.7 | 3.3 | 28.8 | 35.6 | 48.5 |
| MRKLD [9] | 91.0 | 55.4 | 80.0 | 33.7 | 21.4 | 37.3 | 32.9 | 24.5 | 85.0 | 34.1 | 80.8 | 57.7 | 24.6 | 84.1 | 27.8 | 30.1 | 26.9 | 26.0 | 42.3 | 47.1 |
| DT [39] | 90.6 | 44.7 | 84.8 | 34.3 | 28.7 | 31.6 | 35.0 | 37.6 | 84.7 | 43.3 | 85.3 | 57.0 | 31.5 | 83.8 | 42.6 | 48.5 | 1.9 | 30.4 | 39.0 | 49.2 |
| FDA [40] | 92.5 | 53.3 | 82.4 | 26.5 | 27.6 | 36.4 | 40.6 | 38.9 | 82.3 | 39.8 | 78.0 | 62.6 | 34.4 | 84.9 | 34.1 | 53.1 | 16.9 | 27.7 | 46.4 | 50.5 |
| Uncertainty [41] | 90.4 | 31.2 | 85.1 | 36.9 | 25.6 | 37.5 | 48.8 | 48.5 | 85.3 | 34.8 | 81.1 | 64.4 | 36.8 | 86.3 | 34.9 | 52.2 | 1.7 | 29.0 | 44.6 | 50.3 |
| Adaboost [42] | 90.7 | 35.9 | 85.7 | 40.1 | 27.8 | 39.0 | 49.0 | 48.4 | 85.9 | 35.1 | 85.1 | 63.1 | 34.4 | 86.8 | 38.3 | 49.5 | 0.2 | 26.5 | 45.3 | 50.9 |
| FADA [43] | 91.0 | 50.6 | 86.0 | 43.4 | 29.8 | 36.8 | 43.4 | 25.0 | 86.8 | 38.3 | 87.4 | 64.0 | 38.0 | 85.2 | 31.6 | 46.1 | 6.5 | 25.4 | 37.1 | 50.1 |
| CorDA [17] | 94.7 | 63.1 | 87.6 | 30.7 | 40.6 | 40.2 | 47.8 | 51.6 | 87.6 | 47.0 | 89.7 | 66.7 | 35.9 | 90.2 | 48.9 | 57.5 | 0.0 | 39.8 | 56.0 | 56.6 |
| DACS [44] | 89.9 | 39.7 | 87.9 | 30.7 | 39.5 | 38.5 | 46.4 | 52.8 | 88.0 | 44.0 | 88.8 | 67.2 | 35.8 | 84.5 | 45.7 | 50.2 | 0.0 | 27.3 | 34.0 | 52.1 |
| ProDA [45] | 87.8 | 56.0 | 79.7 | 46.3 | 44.8 | 45.6 | 53.5 | 53.5 | 88.6 | 45.2 | 82.1 | 70.7 | 39.2 | 88.8 | 45.5 | 59.4 | 1.0 | 48.9 | 56.4 | 57.5 |
| CaCo [4] | 93.8 | 64.1 | 85.7 | 43.7 | 42.2 | 46.1 | 50.1 | 54.0 | 88.7 | 47.0 | 86.5 | 68.1 | 2.9 | 88.0 | 43.4 | 60.1 | 31.5 | 46.1 | 60.9 | 58.0 |
| BAPA [46] | 94.4 | 61.0 | 88.0 | 26.8 | 39.9 | 38.3 | 46.1 | 55.3 | 87.8 | 46.1 | 89.4 | 68.8 | 40.0 | 90.2 | 60.4 | 59.0 | 0.0 | 45.1 | 54.2 | 57.4 |
| DAFormer [47] | 95.7 | 71.2 | 89.4 | 53.5 | 48.1 | 49.6 | 55.8 | 59.4 | 89.9 | 47.9 | 92.5 | 72.2 | 44.7 | 92.3 | 74.5 | 78.2 | 65.1 | 55.9 | 61.8 | 68.3 |
| DAFormer [47] + CDEA | 97.5 | 76.8 | 92.7 | 58.5 | **58.1** | 52.6 | 65.8 | 69.4 | **92.8** | 49.4 | 94.3 | 74.1 | 48.7 | **96.7** | 77.5 | 81.4 | 67.2 | 59.9 | 63.8 | 72.5 |
| HRDA [23] | 96.4 | 74.4 | 91.0 | 61.6 | 51.5 | 57.1 | 63.9 | 69.3 | 91.3 | 48.4 | 94.2 | 79.0 | 52.9 | 93.9 | 84.1 | 85.7 | 75.9 | 63.9 | 67.5 | 73.8 |
| HRDA [23] + CDEA | **97.8** | **76.9** | **93.0** | **63.2** | 57.9 | **59.4** | **68.9** | **74.3** | 90.9 | **50.3** | **95.6** | **80.5** | **54.9** | 95.7 | **88.2** | **87.3** | 78.2 | **67.7** | **69.4** | **76.3** |

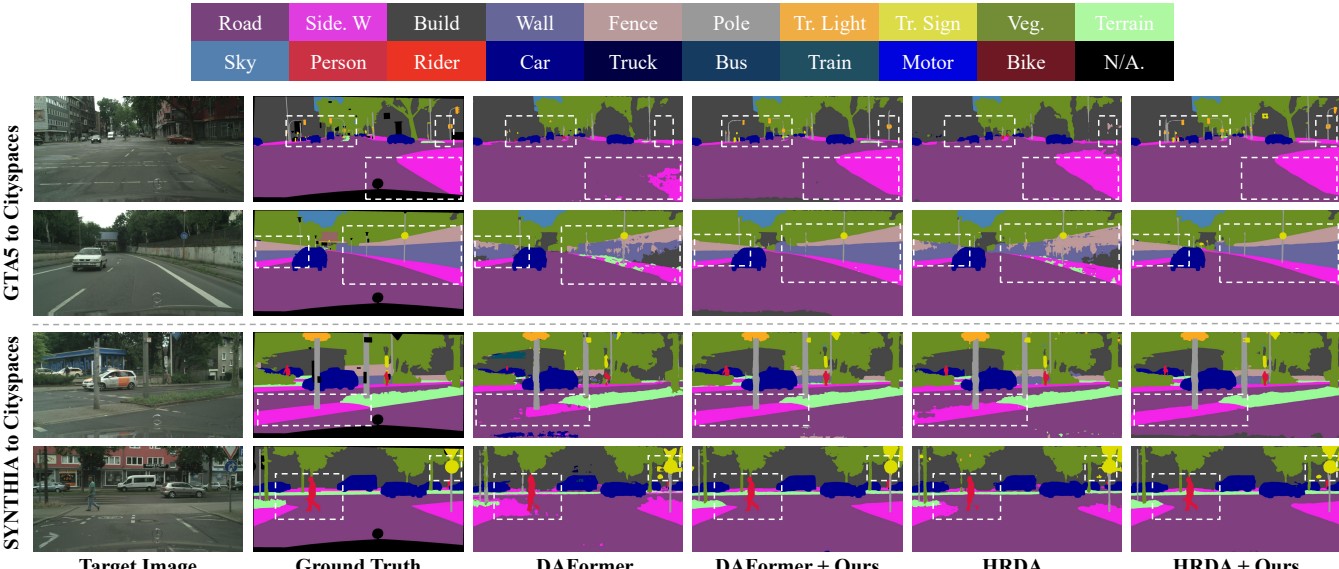

**Figure 3: Qualitative results on GTA → Cityscapes and SYNTHIA → Cityscapes. From left to right: Target Image, Ground Truth, the visual results predicted by DAFormer [47], DAFormer [47] + Ours (CDEA), HRDA [23], HRDA [23] + Ours (CDEA). We deploy the white dash boxes to highlight different prediction parts.**

## 4 Experiments

### 4.1 Implementation Details

We train the network with batch size 2 for 60k iterations with a single NVIDIA Tesla 4*V100 GPU server. We adopt AdamW [51] as the optimizer, a learning rate of $6 \times 10^{-5}$, a linear learning rate warmup of 1.5k iterations, and a weight decay of 0.01. The input image is resized to $1280 \times 720$ for GTA and $1280 \times 760$ for SYNTHIA, with a random crop size of $640 \times 640$. We adopt various data augmentation during training, such as ClassMix [52], Gaussian blur and color jitter. We design CDEA based on a DAFormer network [47]. The teacher network is randomly initialized and the student network is

**Table 2: Quantitative results with previous UDA methods on SYNTHIA → Cityscapes. We present pre-class IoU, mIoU and mIoU*. mIoU and mIoU* are averaged over 16 and 13 categories, respectively. The best accuracy in every column is in bold.**

| Method | Road | SW | Build | Wall* | Fence* | Pole* | TL | TS | Veg. | Sky | PR | Rider | Car | Bus | Motor | Bike | mIoU* | mIoU |
|---|---|---|---|---|---|---|---|---|---|---|---|---|---|---|---|---|---|---|
| SIBAN [48] | 82.5 | 24.0 | 79.4 | – | – | – | 16.5 | 12.7 | 79.2 | 82.8 | 58.3 | 18.0 | 79.3 | 25.3 | 17.6 | 25.9 | 46.3 | – |
| MaxSquare [34] | 77.4 | 34.0 | 78.7 | 5.6 | 0.2 | 27.7 | 5.8 | 9.8 | 80.7 | 83.2 | 58.5 | 20.5 | 74.1 | 32.1 | 11.0 | 29.9 | 45.8 | 39.3 |
| AdaptSegNet [32] | 84.3 | 42.7 | 77.5 | – | – | – | 4.7 | 7.0 | 77.9 | 82.5 | 54.3 | 21.0 | 72.3 | 32.2 | 18.9 | 32.3 | 46.7 | – |
| PatchAlign [37] | 82.4 | 38.0 | 78.6 | 8.7 | 0.6 | 26.0 | 3.9 | 11.1 | 75.5 | 84.6 | 53.5 | 21.6 | 71.4 | 32.6 | 19.3 | 31.7 | 46.5 | 40.0 |
| ASA [33] | 89.2 | 49.5 | 80.4 | 3.7 | 0.3 | 21.7 | 5.5 | 5.2 | 79.5 | 83.6 | 56.4 | 21.0 | 80.3 | 36.2 | 20.0 | 32.9 | 49.3 | 41.7 |
| SP-Adv [7] | 84.8 | 35.8 | 78.6 | – | – | – | 6.2 | 15.6 | 80.5 | 82.0 | 66.5 | 22.7 | 74.3 | 34.1 | 19.2 | 27.3 | 48.3 | – |
| CLAN [31] | 81.3 | 37.0 | 80.1 | – | – | – | 16.1 | 13.7 | 78.2 | 81.5 | 53.4 | 21.2 | 73.0 | 32.9 | 22.6 | 30.7 | 47.8 | – |
| AdvEnt [29] | 85.6 | 42.2 | 79.7 | 8.7 | 0.4 | 25.9 | 5.4 | 8.1 | 80.4 | 84.1 | 57.9 | 23.8 | 73.3 | 36.4 | 14.2 | 33.0 | 48.0 | 41.2 |
| MRNet [35] | 82.0 | 36.5 | 80.4 | 4.2 | 0.4 | 33.7 | 18.0 | 13.4 | 81.1 | 80.8 | 61.3 | 21.7 | 84.4 | 32.4 | 14.8 | 45.7 | 50.2 | 43.2 |
| BL [38] | 86.0 | 46.7 | 80.3 | – | – | – | 14.1 | 11.6 | 79.2 | 81.3 | 54.1 | 27.9 | 73.7 | 42.2 | 25.7 | 45.3 | 51.4 | – |
| CBST [36] | 68.0 | 29.9 | 76.3 | 10.8 | 1.4 | 33.9 | 22.8 | 29.5 | 77.6 | 78.3 | 60.6 | 28.3 | 81.6 | 23.5 | 18.8 | 39.8 | 48.9 | 42.6 |
| CCM [49] | 79.6 | 36.4 | 80.6 | 13.3 | 0.3 | 25.5 | 22.4 | 14.9 | 81.8 | 77.4 | 56.8 | 25.9 | 80.7 | 45.3 | 29.9 | 52.0 | 52.9 | 45.2 |
| MRKLD [9] | 67.7 | 32.2 | 73.9 | 10.7 | 1.6 | 37.4 | 22.2 | 31.2 | 80.8 | 80.5 | 60.8 | 29.1 | 82.8 | 25.0 | 19.4 | 45.3 | 50.1 | 43.8 |
| DADA [50] | 89.2 | 44.8 | 81.4 | 6.8 | 0.3 | 26.2 | 8.6 | 11.1 | 81.8 | 84.0 | 54.7 | 19.3 | 79.7 | 40.7 | 14.0 | 38.8 | 49.8 | 42.6 |
| Uncertainty [41] | 87.6 | 41.9 | 83.1 | 14.7 | 1.7 | 36.2 | 31.3 | 19.9 | 81.6 | 80.6 | 63.0 | 21.8 | 86.2 | 40.7 | 23.6 | 53.1 | 54.9 | 47.9 |
| APODA [6] | 86.4 | 41.3 | 79.3 | – | – | – | 22.6 | 17.3 | 80.3 | 81.6 | 56.9 | 21.0 | 84.1 | 49.1 | 24.6 | 45.7 | 53.1 | – |
| DT [39] | 83.0 | 44.0 | 80.3 | – | – | – | 17.1 | 15.8 | 80.5 | 81.8 | 59.9 | 33.1 | 70.2 | 37.3 | 28.5 | 45.8 | 52.1 | – |
| Adaboost [42] | 85.6 | 43.9 | 83.9 | 19.2 | 1.7 | 38.0 | 37.9 | 19.6 | 85.5 | 88.4 | 64.1 | 25.7 | 86.6 | 43.9 | 31.2 | 51.3 | 57.5 | 50.4 |
| DAFormer [47] | 84.5 | 40.7 | 88.4 | 41.5 | 6.5 | 50.0 | 55.0 | 54.6 | 86.0 | 89.8 | 73.2 | 48.2 | 87.2 | 53.2 | 53.9 | 61.7 | 67.4 | 60.9 |
| DAFormer [47] + CDEA | 86.7 | 42.9 | 89.7 | 42.7 | **8.1** | 51.4 | 57.3 | 56.2 | **88.3** | 91.2 | 75.1 | 49.8 | **90.1** | 56.4 | 55.7 | 63.3 | 69.4 | 62.8 |
| HRDA [23] | 85.2 | 47.7 | 88.8 | 49.5 | 4.8 | 57.2 | 65.7 | 60.9 | 85.3 | 92.9 | 79.4 | 52.8 | 89.0 | 64.7 | 63.9 | 64.9 | 72.4 | 65.8 |
| HRDA [23] + CDEA | **88.6** | **54.8** | **90.6** | **52.3** | 7.4 | **62.4** | **68.8** | **64.6** | 87.5 | **93.9** | **83.6** | **56.6** | 90.0 | **58.7** | **66.6** | **68.5** | **74.8** | **68.4** |

pre-trained on ImageNet1k. The exponential moving average parameter $\alpha$ of the teacher network is 0.999 [13]. The hyperparameters of the loss function are chosen empirically $\lambda^T = 2, \lambda^M = 1, \lambda^D = 1$ and the ratio of pixels exceeding a threshold $\tau = 0.5$. We report our results for 19 classes on GTA→Cityscapes and both 13 and 16 classes on SYNTHIA→Cityscapes.

## 4.2 Datasets and Evaluation Metrics

**Datasets.** In this paper, we use three commonly used datasets in UDA. **GTA5** is a synthetic dataset containing 24,966 high-resolution images collected from game video, and the corresponding ground-truth segmentation map can be generated by computer graphics. **SYNTHIA** is also a synthetic dataset, which contains 9,400 images. It shares 16 common classes with Cityscapes dataset. **Cityscapes** is a real-world dataset collected for autonomous driving scenarios from 50 cities around the world. It contains 2,975 and 500 images for training and validation, respectively.

**Evaluation Metrics.** For segmentation evaluation, we adopt the mean of class-wise intersection over union (mIoU) as the evaluation metrics.

**Reproducibility.** The code is based on Pytorch and MMSegmentation. We will make our code open-source for reproducing all results.

## 4.3 Comparisons with State-of-the-art Methods

We compare CDEA with several competitive UDA methods on GTA → Cityscapes and SYNTHIA → Cityscapes, respectively. The quantitative comparisons are shown in Tab. 1 and Tab. 2, respectively. We highlight the best accuracy in every column in bold. Meanwhile, we visualize the visual result between the proposed method and the other two state-of-the-art Transformer methods [23, 47] in Fig. 3.

**GTA → Cityscapes.** We visualize all experiment results on GTA → Cityscapes in Tab. 1. It could be observed that Transformer methods significantly surpass the CNN methods by a large margin since

DAFormer [47]. The results show that our CDEA performed a remarkable improvement over the existing SOTA transformer models DAFormer [47] and HRDA [23]. Meanwhile, CDEA achieves 72.5 mIoU, which outperforms DAFormer [47] by a considerable margin of +4.2 mIoU. In addition, we improve +2.5 mIoU and achieve the SOTA performance of 76.3 mIoU, verifying the effectiveness of the CDEA that introduces a unified and multi-grained unsupervised learning algorithm in UDA task, when applying CDEA to HRDA [23] Moreover, CDEA achieves leading IoU of almost all classes on GTA → Cityscapes, including several small-object categories such as "fence", "pole", "training light" and "training sign".

**SYNTHIA → Cityscapes.** As shown in Tab. 2, CDEA also achieves significant mIoU and mIoU* (13 most common categories) performance on SYNTHIA → Cityscapes, increasing +1.9 and +2.6 mIoU compared with DAFormer [47] and HRDA [23], respectively. It is noticeable that our CDEA remains competitive in segmenting each individual class including small-scale objectives and yields the best IoU score in almost all categories. The IoU performance of CDEA verifies our innovation that the exploration of the inherent structures of intra-domain images indeed helps category recognition, especially for challenging small objects.

**Qualitative Results.** As shown in Fig. 3, we show the segmentation prediction results of CDEA and the comparison with existing SOTA methods DAFormer [47], HRDA [23], and the original image with ground truth on both GTA → Cityscapes and SYNTHIA → Cityscapes benchmarks. The results show that CDEA can segment the detail of edge and minor categories, such as "fence", "wall", "traffic sign" and "traffic light", which are highlighted by white bounding boxes. It is also noticeable that CDEA predicts smoother edges between different categories, "traffic sign" and "person" in the fourth row of Fig. 3. The reason is that the proposed detail-enhanced

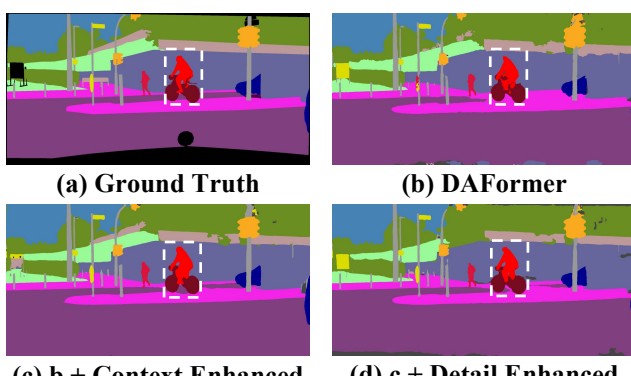

| (a) Ground Truth | (b) DAFormer |
|:---:|:---:|
| (c) b + Context Enhanced | (d) c + Detail Enhanced |

**Figure 4: Effect of Context and Detail-Enhanced**

**Table 3: Effect of the patch Mask Ratio and Size on GTA → Cityscapes.**

| No. | Type | Mask Ratio | Mask Size | mIoU |
|:---:|:---:|:---:|:---:|:---:|
| 1 | | 0.3 | $32 \times 32$ | 70.4 |
| 2 | Fixed | 0.6 | $64 \times 64$ | 71.0 |
| 3 | | 0.5 | $128 \times 128$ | **71.3** |
| 4 | | 0.4 | $256 \times 256$ | 70.8 |
| 5 | | 0.2~0.7 | 32~128 | 71.9 |
| 6 | Random | 0.3~0.8 | 64~256 | 72.1 |
| 7 | | 0.2~0.7 | 32~128 | **72.5** |
| 8 | | 0.3~0.8 | 64~256 | 72.2 |

**Table 4: Effect of the Detail of UpSample and DownSample on GTA → Cityscapes.**

| No. | Up/Down 1x | 2x | 4x | mIoU | No. | Up/Down 1x | 2x | 4x | mIoU |
|:---:|:---:|:---:|:---:|:---:|:---:|:---:|:---:|:---:|:---:|
| 1 | ✓ | | | 70.4 | 4 | ✓ | ✓ | | 71.2 |
| 2 | | ✓ | | 71.1 | 5 | | ✓ | ✓ | 71.5 |
| 3 | | | ✓ | 70.8 | 6 | ✓ | ✓ | ✓ | **72.5** |

module explicitly encourages detail-wise consistency against different contexts, which improves the prediction robustness on the edges of categories.

## 4.4 Ablation Studies and Further Analysis

In this section, we introduce the ablation experiments to validate the effectiveness of each component in our method. We verify the effectiveness of the two proposed components, Context Guidance and Detail Guidance in the proposed CDEA and investigate how set the parameter of two Guidance contributes to the final performance on GTA → Cityscapes. For a fair comparison, we apply the same experimental environment and data.

**Effectiveness of Context Guidance.** For context guidance, the ratio and size of the mask also affect performance and training difficulty. We evaluated the performance of both fixed and random types. As shown in Tab. 3, we could observe: The best performance is 72.5

**Table 5: Ablation study on the effect of Context and Detail-Enhanced on GTA → Cityscapes.**

| Method | $\mathcal{L}_{\text{Context}}$ | $\mathcal{L}_{\text{Detail}}$ | mIoU | ΔmIoU |
|:---|:---:|:---:|:---:|:---:|
| DAFormer [47] | | | 68.3 | − |
| +Context Enhanced | ✓ | | 69.5 | +1.2 |
| +Detail Enhanced | | ✓ | 71.4 | +3.1 |
| +CDEA | ✓ | ✓ | **72.5** | +4.3 |

mIoU when the mask size range is 32~128 and the mask ratio range is 0.2~0.7 by random type. The best performance of fixed type is 71.3 mIoU when the mask size is 128×128 and the mask ratio is 0.5 by fixed type. The experimental results show that the random type is obviously better than the fixed method, indicating that the random type can improve context awareness and accuracy by random mask.
**Effectiveness of Detail Guidance.** For detailed guidance, The different scales of Up/DownSample also affect the accuracy and performance [12]. As shown in Tab. 4, we gradually utilize and combine the different scales of Up/DownSample. We observe that larger patch generally obtain better performance since it contains more diverse contexts and is close to the test size. The best performance is 72.5 mIoU when the mask size range is 32~128 and the mask ratio range is 0.2~0.7 by random type. The experimental results demonstrate the effectiveness of Up/DownSample with multiple scales.
**Effectiveness of the Both Guidance.** The baseline model is based on DAFormer [47], which outputs a competitive mIoU of 68.3, shown in Tab. 5 When applying detailed guidance and context guidance individually could lead to +1.2 mIoU, +3.1 and +4.3 mIoU improvement respectively, verifying the effectiveness of exploring the inherent contextual information, shown in Tab. 5. When applying both guidances, our CDEA further improves the performance to 72.5 mIoU, surpassing the model that deploys only one kind of guidance. The experimental results visually demonstrate the effectiveness of the method, shown in Fig. 4. CDEA effectually learns the multi-level information by combining the two kinds of guidance. Therefore, the two kinds of guidance are complementary to each other.

## 5 Conclusion

In this paper, we presented Context- and Detail-Enhanced UDA method (CDEA) to improve the learning of target domain context relations and detail feature. We target to learn a feature space that enables discriminative context features and the robust feature learning of the detail feature against variant contexts. CDEA can promote the model to extract the inherent contextual and detail feature, which is domain invariant. In comprehensive experiments, we have shown that CDEA achieves significant performance improvements in all of these UDA tasks. For instance, CDEA respectively improves the SOTA performance by +2.5 and +2.6 on GTA→Cityscapes and SYNTHIA → Cityscapes. Since the simplicity of CDEA, it can be combined with other existing methods to further facilitate the intra-domain knowledge learning. In addition, We hope that CDEA can be used as part of future UDA methods to narrow the gap between UDA learning.

## Acknowledgments

This work was supported by the Outstanding Research Project of
Shen Yuan Honors College, BUAA, under Grant 230123104.

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
