# OpenReview forum: "CDEA: Context- and Detail-Enhanced Unsupervised Learning for Domain Adaptive Semantic Segmentation"
_acmmm.org/ACMMM/2024/Conference — MM2024 Poster_

### Official Review · Reviewer_PdE4 · 2024-05-04

**Rating:** 3
**Confidence:** 4

**Summary:**

This paper introduces an innovative approach to enhance unsupervised domain adaptive semantic segmentation by incorporating spatial context relations as supplementary information. The study proposes a context- and detail-enhanced unsupervised learning framework designed to leverage both context- and detail-wise consistency across varying contexts. This framework aims to improve the robustness and accuracy of semantic segmentation models by effectively utilizing spatial context relations to better understand and adapt to new, unseen domains. Extensive experiments are conducted on two popular datasets to validate the effectiveness of the proposed method. The results demonstrate significant improvements in model performance, highlighting the potential of spatial context relations in enhancing domain adaptability and segmentation precision in unsupervised settings.

**Strengths:**

1. The figures included in the manuscript are of high quality and the writing is clear and well-structured.

2. Using a spatial context relation to improve the accuracy is interesting.

3. The paper also presents good ablation studies and reports competitive experimental results to support the claims.

**Limitations:**

1. The template used by the author is incorrect; the layout and related content need to be checked.  Like " Conference’17, July 2017, Washington, DC, USA " is wrong.

2. How does the proposed method address the issue of "similar local looking" by utilizing spatial context relations? Many other works [1][2] also utilize context- and detail-wise consistency to address this issue. What sets this paper's approach apart from others?

[1] MIC: Masked Image Consistency for Context-Enhanced Domain Adaptation.
[2] Cross-level Contrastive Learning and Consistency Constraint for Semi-Supervised Medical Image Segmentation.

3. Is the problem shown in Fig. 1(a) solved via more spatial context relations? How does spatial context knowledge solve the problem of similar local looking? Is that solved due to the introduction of relative positions and relationships in the scene?

4. Recent method also enhances the context understanding via multi-modal fusion of visual and depth knowledge, the difference and similarity between CDEA and [3] should be discussed.
[3] Transferring to Real-World Layouts: A Depth-aware Framework for Scene Adaptation.

5.  How do you choose different weight values in the total loss function (Eqn. 12)? Is there any sensitivity analysis?

**Suitability:**

3

---

### Official Review · Reviewer_H3mS · 2024-05-16

**Rating:** 6
**Confidence:** 2

**Summary:**

This paper proposes an unsupervised learning framework that integrates context and detail enhancement. On one hand, the study introduces an adaptive masked image consistency module to learn spatial context information in the target domain, thereby improving the performance of UDA. On the other hand, the study proposes a detail extraction module that integrates the learning of spatial information at lower levels with deep semantic features. Additionally, the proposed method can be effectively embedded into other models. The method has been validated across multiple datasets, demonstrating strong competitiveness and providing a powerful tool for semantic segmentation tasks.

**Strengths:**

Novelty: This paper propose a context and detail-enhanced unsupervised learning framework to harness both context- and detail-wise consistency against different contexts, which is well-aligned with the segmentation task.
Theoretical Approach: Utilizing spatial contextual information and detail information to improve the performance of semantic segmentation may be theoretically reasonable.
Evaluation: The performance improvement is demonstrated through comparisons with existing methods on two sets of datasets, highlighting the effectiveness of the proposed method.
Clarity: The structure of the paper is relatively clear.

**Limitations:**

1.Please code the references in order of citation, as done with reference 40 in the introduction.
2.In Section 3.2, right column, ". The domain adaptation process with the CEM is illustrated in Fig. 3 and explained below." There is an incorrect citation of Fig.3.
3.In Section 3.2, right column, the meaning of the letter μ in formula (7) is not explained, resulting in an error in the description.
4.Please add the display and analysis of failure cases.

**Suitability:**

3

---

### Official Review · Reviewer_iHgY · 2024-05-23

**Rating:** 4
**Confidence:** 4

**Summary:**

This paper introduces a context- and detail-enhanced unsupervised learning framework (CDEA) for domain adaptive semantic segmentation. In this framework, it involves Context Enhancement and Detail Enhancement. For Context Enhancement,  it proposes an adaptive masked image consistency module to improve UDA by learning spatial context relations in the target domain, ensuring consistency between predictions and masked target images. For Detail Enhancement, it introduces a detail extraction module that integrates spatial information learning into low-level layers, fusing low-level detail features with deep semantic features.

**Strengths:**

1. The paper is well-written with thorough attention to detail.
2. While the contributions of the paper may not be entirely novel, the authors address practical concerns and provide extensive design considerations.
3. The experiments are comprehensive, demonstrating significant advantages in performance.

**Limitations:**

Certainly, I don't have many concerns. I rated the paper as Borderline because the concepts of mask and low-level detail used in the paper are not entirely novel, but I can appreciate incremental innovation. This isn't my main issue, so there's no need for the authors to defend themselves. Here are my questions:

1. While the detail provided by the authors resembles edge maps, have they attempted to directly use edge extraction as ground truth for supervision?
2. Have the authors considered trying multiple masking approaches and conducting ablation experiments to determine the best mask?
3. Is it possible that the symbols for predicted detail and label detail in Figure 2 are reversed?

**Suitability:**

2

---

### Official Review · Reviewer_EPTL · 2024-05-24

**Rating:** 2
**Confidence:** 4

**Summary:**

This paper studies Unsupervised Domain Adaptation (UDA) in semantic segmentation and introduces CDEA, a Context- and Detail-Enhanced Unsupervised Learning framework. CDEA improves UDA by firstly introducing an Adaptive Masked Image Consistency module, which learns spatial context relations in the target domain, enforcing consistency between predictions and masked target images. Secondly, a Detail Extraction module integrates spatial information into low-level layers, combining low-level detail features with deep semantic features. Extensive experiments validate the effectiveness of CDEA, showcasing its superiority over state-of-the-art methods.

**Strengths:**

1. The paper is well organized, the figures are readable and understandable.
2. The proposed method looks logical and technically sound.
3. The experimental results are good. Consistent improvements have been shown in different benchmarks.

**Limitations:**

1. Concerns on the novelty: Previous work (MIC [1]) has also explored mask image consistency between predictions and masked target images. Directly applying a similar idea seems does not introduce sufficient novelties. Please the authors make extensive discussions on the differences.

[1]. MIC: Masked Image Consistency for Context-Enhanced Domain Adaptation, CVPR 2023;

2. Insufficient comparison results:  1) The authors only conduct experimental comparisons on synthetic-to-real adaptation. Existing papers [1-2] also perform Day-to-Nighttime adaptation (Cityscapes→DarkZurich) and Clear-to-Adverse-Weather adaptation (Cityscapes→ACDC). The reviewer wonders whether are there any results on such benchmarks that could verify the effectiveness of the proposed method. 2) The authors only verify the effectiveness of CDEA using DAFormer and HRDA on domain adaptation benchmarks. In addition to them does the presented method also work on the MIC [1]? What are the improvements on the superior baseline of MIC[1]?

[2]. Domain Adaptive and Generalizable Network Architectures and Training Strategies for Semantic Image Segmentation, IEEE TPAMI 2023;



3. Insufficient experimental analysis: There's a lack of comprehensive experimental analysis, including TSNE feature visualization, visualization of source and target domains, class visualization, and the distribution of high-level and low-level features. These analyses are crucial for understanding the model's behavior and performance, identifying areas for improvement, and assessing how the model performs in different environments.

4. Lack of citations. Some important references [3-11] that also target context-aware domain adaptation [3-6] and Mean Teacher architecture for domain adaptive semantic segmentation [7-11] are missing. These works should be briefly reviewed in the related works. In addition, some recent works should be compared in tables.

[3]. Context-Aware Domain Adaptation in Semantic Segmentation, WACV 2021;

[4]. Domain Adaptive Semantic Segmentation with Regional Contrastive Consistency Regularization. ICME 2022;

[5]. Context-Aware Pseudo-label Refinement for Source-Free Domain Adaptive Fundus Image Segmentation, MICCAI 2023;

[6]. Contextual-Relation Consistent Domain Adaptation for Semantic Segmentation, ECCV 2020;

[7].Self-Ensembling Attention Networks: Addressing Domain Shift for Semantic Segmentation. AAAI 2019;

[8]. Uncertainty-Aware Source-Free Domain Adaptive Semantic Segmentation, IEEE TIP 2023;

[9]. Uncertainty-Aware Consistency Regularization for Cross-Domain Semantic Segmentation. CVIU 2022;

[10]. Towards Better Stability and Adaptability: Improve Online Self-Training for Model Adaptation in Semantic Segmentation, CVPR 2023;

[11]. DiGA: Distil to Generalize and then Adapt for Domain Adaptive Semantic Segmentation, CVPR 2023;

**Suitability:**

2

---

### Meta-Review · Area_Chair_hqDJ · 2024-06-25

**Recommendation:** Accept (Poster)
**Confidence:** 5

**Metareview:**

All reviewers give positive suggestions.

Using a spatial context relation to improve the accuracy is interesting.

The paper also presents good ablation studies and reports competitive experimental results to support the claims.